# ChinaWheatYield30m: A 30-m annual winter wheat yield dataset from 2016 to 2021 in China

Yu Zhao[1,2#], Shaoyu Han[1,3#], Jie Zheng[1], Hanyu Xue[1], Zhenhai Li[1,4], Yang Meng[1,2], Xuguang Li[5], Xiaodong Yang[1], Zhenhong Li[6], Shuhong Cai[5], Guijun Yang[1,6*]

[1] Key Laboratory of Quantitative Remote Sensing in Agriculture of Ministry of Agriculture and Rural Affairs, Information Technology Research Center, Beijing Academy of Agriculture and Forestry Sciences, Beijing 100097, China
[2] College of agriculture, Shanxi Agricultural University, Taigu, Shanxi 030801, China
[3] Institute of Agricultural Economics and Information, Henan Academy of Agricultural Sciences, Zhengzhou, Henan 450000, China
[4] College of Geodesy and Geomatics, Shandong University of Science and Technology, Qingdao 266590, China
[5] Cultivated Land Monitoring and Protection Center of Hebei, Shijiazhuang, 050056, China
[6] School of Geological Engineering and Geomatics, Chang'an University, Xi'an 710054, China

[#] these authors contributed equally as first authors

*Correspondence to*: Guijun. Yang (guijun.yang@163.com)

**Abstract.** Generating spatial crop yield information is of great significance for academic research and guiding agricultural policy. Existing public yield datasets have a coarse spatial resolution, spanning from 1 km to 43 km. Although these datasets are useful for analyzing large-scale temporal and spatial change in yield and they cannot deal with small-scale spatial heterogeneity, which happens to be the most significant characteristic of the Chinese farmers' economy. Hence, we generated a 30-m Chinese winter wheat yield dataset (ChinaWheatYield30m) for major winter wheat-producing provinces in China for the period 2016-2021 with a semi-mechanistic model (hierarchical linear model, HLM). The yield prediction model was built by considering the wheat growth status and climatic factors. It can estimate wheat yield with excellent accuracy and low cost using a combination of satellite observations and regional meteorological information (i.e., Landsat 8, Sentinel 2 and ERA5 data from the Google Earth Engine (GEE) platform). The results were validated by using in situ measurements and census statistics and indicated a stable performance of the HLM model based on calibration datasets across China, with r of 0.81 and rRMSE of 12.59%. With regards to validation, the ChinaWheatYield30m dataset was highly consistent with in situ measurement data and statistical data (p < 0.01), indicated by r (rRMSE) of 0.72** (15.34%) and 0.69** (19.16%). The ChinaWheatYield30m is a sophisticated dataset with both high spatial resolution and convicting accuracy, such a dataset will provide basic knowledge of detailed wheat yield distribution, which can be applied for many purposes including crop production modelling or regional climate evaluation.

## 1 Introduction

Wheat is the most widely planted crop, supplying a fifth of global food calories and protein (Erenstein et al., 2022). However, wheat production is facing unprecedented challenges in the global context of climate change, such as frequent extreme weather events. Apart from natural factors, socioeconomic events such

as the COVID-19 pandemic, regional conflicts, and other global crises can also significantly perturb wheat production (IFPRI, 2022). In China, where needs to feed one-fifth of the world's population on its limited land (FAO, 2020) and food security is crucial, wheat production is an essential agricultural activity. Ensuring stable grain supplies and increasing production are important to the national economy and people's livelihoods (Feng et al., 2020). Therefore, monitoring of crop yields timely is of great

significance for regulating import and export decision-making, grain market prices, crop insurance evaluations, smart agriculture applications, and rational allocations of agricultural resources.

In the past decades, remote sensing data from ground-based, aerial-based and satellite-based platforms have received extensive attention for crop yield prediction (Battude et al., 2016; Jiang et al., 2019; Li et al., 2020; Wang et al., 2021). Ground- and aerial-based platforms have high spatial resolution and control,

which are advantageous for farm-scale applications. However, their application to large-area yield estimations is too expensive. Satellite-based approaches have been widely used to monitor crop production over large areas in the past few decades, benefitting from capable of acquiring temporally and spatially continuous information (Battude et al., 2016; Huang et al., 2019). With the rapid launch of new satellites carrying various types of sensors, regional yield mapping is becoming more accurate and at

higher spatial resolution. The mapping relies on vegetation indices (VIs) that can be derived from visible and near-infrared (NIR) reflectance bands in multispectral optical data, such as the Normalized Difference Vegetation Index (NDVI) (Rouse et al., 1974), the enhanced vegetation index (EVI) (Sims et al., 2008), or the optimized soil adjust vegetation index (OSAVI) (Rondeaux et al., 1996). These VIs have often been used to predict crop yield (Magney et al., 2016; Cao et al., 2021; Zhao et al., 2022b).

There are many methods to incorporate VIs in yield estimation, such as parametric regressions, deep learning, and data assimilation (Battude et al., 2016; Huang et al., 2019; Li et al., 2020).

Parametric regression models directly establish the relationship between VIs and crop yield, which may be linear or nonlinear (Magney et al., 2016; Li et al., 2020). These parametric regressions are limited to the specific research area and growing season for which they are developed, making it hard to extrapolate

them either in the spatial or temporal domains. Non-parametric statistical approaches have been used in recent yield projections research. Notable studies have been done using machine learning (ML) (Cai et al., 2019; Li et al., 2021). An emerging new technique for crop yield estimations is deep learning (Tian et al., 2021) applied to various types of data acquired by satellites and drones (Jiang et al., 2020; Wang et al., 2020). Overall, ML methods heavily rely on large training datasets (Cao et al., 2021). Nonetheless,

the application of machine learning in the realm of synthetic data generation has also exhibited encouraging outcomes (Arslan et al., 2019; Sivakumar et al., 2022; Ebrahimy et al., 2023).

Unlike the above-mentioned statistical models, process-based mechanic models simulate crop yield from various inputs, including soil properties, meteorological data as well as crop characters. Examples of such models are the Decision Support System of Agrotechnology Transfer modeling system (DSSAT),

the Agricultural Production Systems sIMulator (APSIM) and the Simple Algorithm For Yield (SAFY) and many other crop models (Jones et al., 2003; Keating et al., 2003; Duchemin et al., 2008). These mechanistic models can generate reliable yield estimates (Paudel et al., 2021). Data assimilation provides a way of integrating the monitoring properties of observed data into the predictive and explanatory abilities of crop growth models. Leaf area index (LAI) or biomass are often used as state variables of the

DA system to correct a crop growth model behavior and ensure accurate yield predictions (Battude et al., 2016; Kang and Ozdogan, 2019). Yield is a complex trait that is related to numerous factors, including natural drivers (Li et al., 2021), crop variety (Wei et al., 2022; Bailey-Serres et al., 2019), and human factors, majorly consisting of fertilization and irrigation (Jones et al., 2003; Keating et al., 2003; Duchemin et al., 2008). Existing studies demonstrated that only updating one or two state variables is

not sufficient to correct a crop growth model and thus cannot improve output predictions (Ines et al., 2013; Huang et al., 2015; Hu et al., 2017; Huang et al., 2019). In addition, uncertainties in the remote sensing monitoring of state variables such as LAI and biomass are also inherited by the DA system (Kang et al., 2019). Although data assimilation techniques allow a formal and well-understood way to combine model predictions with observations, their computational intensity is a problem that tends to be ignored

when estimating large-area crop production. Transfer learning techniques can be used to transfer the knowledge learned from a crop growth model to predict wheat yield to effectively improve calculation efficiency (Zhao et al., 2022b). A reliable labeled dataset is a prerequisite for the transfer learning method (Zhang et al., 2021). However, building an effective dataset for transfer learning over a large region is still challenging.

In addition to traditional crop models and assimilation strategies, there are hybrid models that incorporate the simplicity of a statistical model and the rationality of a mechanistic model and are thus called semi-mechanistic models (Ji et al., 2022). For example, Dong et al. (2020b) developed the EC-LUE-GPP model and successfully estimated the wheat yield in Kansas, USA. Li et al. (2020) used the HLM model to estimate interannual yield and showed good performance. Generally, a semi-mechanistic model has

great potential in yield estimation, but its application is often limited to a relatively small area, e.g., farm scale, county or city scale, rather than a larger scale. National crop yield datasets, which are of great significance for large-scale agricultural resource allocation, agricultural system model construction, and climate change impact assessment, are produced at coarse spatial resolutions (Table 1), e.g., 0.5°, 10 km, 4 km or 1 km resolution (Monfreda et al., 2008; You et al., 2014; Iizumi and Sakai., 2020; Grogan et al.,

2022; Luo et al., 2022; Cheng et al., 2022) and are mostly downscaled based on the statistical yield datasets and other datasets (Monfreda et al., 2008; You et al., 2014; Iizumi and Sakai., 2020; Grogan et al., 2022). This method of yield downscaling may lead to inaccurate yield estimates and incorrect assessments of the impact of climate change. In addition, yield predictions cannot rely on statistical data alone. Luo et al. (2022) and Cheng et al. (2022) developed yield datasets combining coarse-resolution

real-time remote sensing data with agricultural statistics, but because 1 km × 1 km plots or 4 km × 4 km farmlands are rare in China, their field application is limited. Although these datasets are useful for analyzing larger-scale temporal and spatial changes in yield, they cannot deal with small-scale spatial heterogeneity, which happens to be the most significant characteristic of the Chinese farmers' economy. Therefore, there is an urgent need to construct a high-resolution yield dataset for investigating

spatiotemporal patterns of crop production, assessing climate change impacts, and modeling crop growth processes over large spatial extents.

**Table 1 Summary of studies on crop yield datasets**

| Method | Species | Resolution | Span | Spatial coverage | References |
|---|---|---|---|---|---|
| Dataset summary | 175 crops | 10 km | 2000 | Global | Monfreda et al., 2008 |
| Global spatial production allocation model | 20 crops | 10 km | 2000, 2005, 2010 | Global | You et al., 2014 |
| Maize, Rice, Wheat and Soybean | 4 crops | 43 km | 1981-2016 | Global | Iizumi & Sakai, 2020 |
| Gata statistics based on Global Agro-Ecological Zones Version 4 model | 26 crops | 10 km | 2015 | Global | Grogan et al., 2022 |
| LSTM | Wheat | 4km | 1982 - 2020 | Global | Luo et al., 2022 |
| Random Forest | Maize, Wheat | 1 km | 2001 - 2015 | China | Cheng et al., 2022 |

In this study, by integrating remote sensing and climate data, we aim to 1) propose a semi-mechanistic model with excellent accuracy and low cost by combining remote sensing observations and regional meteorological information, which can simultaneously overcome inter-annual and cross-regional problems; 2) evaluate model performance by using both validation dataset and the census yield data; 3) generate a high-resolution Chinese winter wheat yield dataset (ChinaWheatYield30m) for the period 125 2016-2021. This dataset will be useful to further yield-related research and guide related food policies.

**2 Data and methods**

**2.1 Study areas**

Our study area consists of the main winter wheat-growing region of China, which includes 12 provinces and municipalities (Figure. 1). The main winter wheat production areas are mainly distributed in the 130 Huang-Huai-Hai region (HHH), Southwest China (SW), Gansu-Xinjiang region (GX), the middle and lower reaches of the Yangtze River (MLYR), and the Loess Plateau (LP). Most of the region is in the middle of China and includes temperate-continental monsoon, temperate monsoon, and subtropical monsoon climates. The sown area and production of winter wheat in China accounted for 20.02% and

21.77% of staple food crops in 2021 (National Bureau of Statistics of China, 2021), respectively. Three sample areas were selected for detailed analysis based on their different geographical and climatic conditions. The three selected regions in this study were chosen for comparison with other yield datasets based on different wheatland coverages. Region 1, 2, and 3 represent areas with winter wheat coverages below 25%, around 50%, and above 75%, respectively, serving as representative regions for these respective coverage levels.

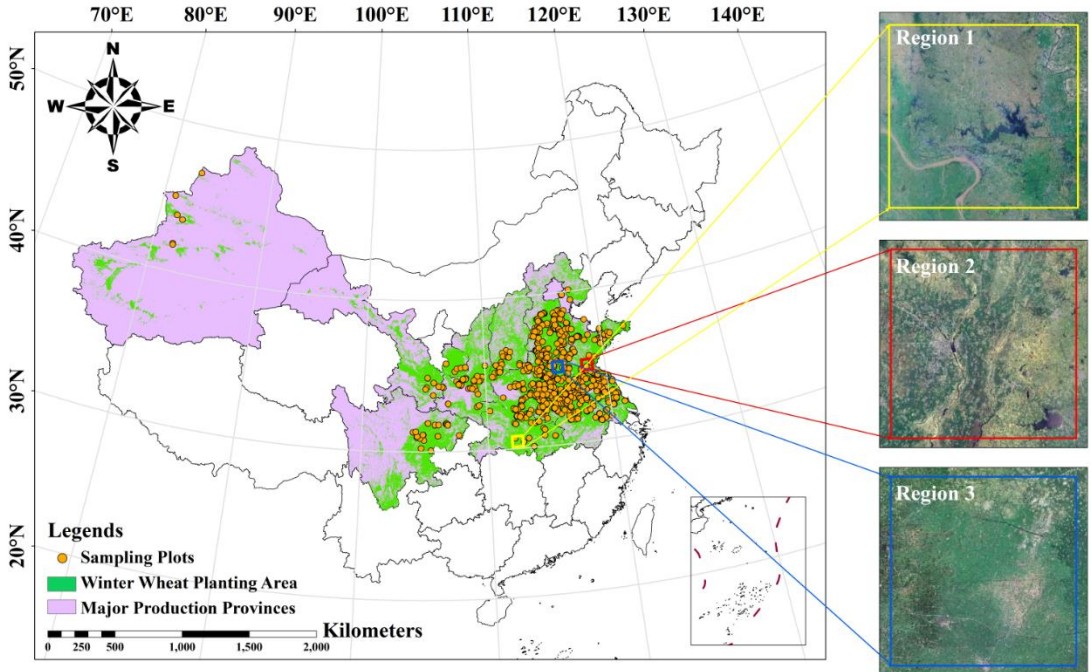

**Figure 1. Distribution of winter wheat within the study area and three selected example areas. Region 1, 2, and 3 represent areas with winter wheat coverages below 25%, around 50%, and above 75%, respectively, serving as representative regions for these respective coverage levels.**

### 2.2 Data Collection

#### 2.2.1 The winter wheat land cover data

We used a winter wheat map with a 30-m resolution across the main growing areas of China (Dong et al., 2020a). These data produce winter wheat maps from 2016 to 2020, which is the base map of ChinaWheatYield30m production. The yield distribution map of 2021 uses the winter wheat classification map of 2020, and the rest of the yield distribution maps are winter wheat classification maps of that year.

#### 2.2.2 Satellite Imagery Data Acquisition

In this work, we extracted the atmospherically corrected reflectance from Landsat 8 and Sentinel 2 images on the Google Earth Engine (GEE) platform during the period of 2016-2021. Subsequently, we calculated the Enhanced Vegetation Index 2 (EVI2) (Jiang et al., 2008) using the extracted reflectance values. These datasets were chosen to increase observation frequency and were used for yield estimation. Xu et al. (2020) have shown that Landsat 8 data and Sentinel 2 data have high consistency. The EVI2 is calculated from the reflectance in Red and NIR bands (Eq. (1)):

$$EVI2 = 2.5 * \frac{NIR - Red}{NIR + 2.4 * Red + 1} \tag{1}$$

where NIR and Red represent the Near-Infrared and Red reflectance, respectively, in Landsat 8 or Sentinel 2. The maximum EVI2 ($EVI2_{max}$) of the winter wheat growing season was used in this paper. It is generally believed that the time of $EVI2_{max}$ corresponds to the heading period, which has been shown to be the best period for remote sensing yield estimation (Luo et al., 2020).

### 2.2.3 Meteorological data

Meteorological data were important input variable for yield prediction, mainly from March to May, because this period includes most key growth stages of winter wheat (i.e. stem elongation, booting, heading, flowering and filling stages). The meteorological data, including monthly average temperatures (Tem), monthly solar radiation (Rad), and monthly precipitation (Pre), were obtained from the ERA5 dataset provided by the GEE platform (https://developers.google.com/earth-engine/datasets/catalog/ECMWF_ERA5_LAND_MONTHLY) with a resolution of 0.1° for the sampling site. All three types of meteorological datasets were resampled to a 30-m resolution to ensure data uniformity.

### 2.2.4 In situ measurement yield data

Georeferenced field-scale yields were obtained by field investigation from 2016 to 2021. During the harvest period, a five-point (1 m$^2$ per point) sampling method was used to destructively sample each winter wheat plot to measure yield. To avoid edge effects, each sample point was at least 2 m away from the edge of the farmland. The harvested grain was threshed and air-dried for yield determination. Then, the final yield was standardized as grain with 14% moisture content. The detailed collection numbers of samples from different regions are shown in Table 2. In this paper, the data were randomly split into two dataset, two-thirds of the data were used for modelling, and the remaining data were used for validation.

**Table 2 Detailed statistics on the sample numbers in this study.**

| Province | Anhui | Gansu | Hebei | Henan | Hubei | Jiangsu | Shaanxi | Shandong | Shanxi | Sichuan | Tianjin | Xinjiang | Total |
|---|---|---|---|---|---|---|---|---|---|---|---|---|---|
| 2016 | 12 | 8 | 26 | 45 | - | 33 | - | 10 | 3 | 11 | 1 | - | 149 |
| 2017 | 53 | 4 | 35 | 72 | 16 | 46 | 25 | 59 | 11 | 9 | 1 | 2 | 333 |
| 2018 | 85 | 3 | 63 | 126 | 18 | 47 | 21 | 56 | 14 | 13 | 1 | 3 | 450 |
| 2019 | 85 | 3 | 48 | 130 | 13 | 53 | 17 | 62 | 14 | 10 | | 2 | 437 |
| 2020 | 82 | 10 | 26 | 121 | 11 | 60 | 19 | 52 | 14 | 0 | - | - | 395 |
| 2021 | 81 | 7 | 25 | 125 | 10 | 26 | 18 | 64 | 8 | 7 | 2 | 3 | 376 |
| Total | 398 | 35 | 223 | 619 | 68 | 265 | 100 | 303 | 64 | 50 | 5 | 10 | 2140 |

Note: "-" represents no collected data.

### 2.2.5 The province-level and municipal-level statistical data

The province-level and municipal-level yield data for the study area were collected from state statistical bureau between 2016 and 2021 (http://www.stats.gov.cn/tjsj/ndsj/). However, the data collected did not have direct records of the unit yield data. Therefore, to obtain the statistical yield data (kg·ha$^{-1}$), the total

production was converted by dividing the planted area. These data were used to validate the model in the selected research provinces and municipalities. Table 3 shows the main information and sources of all data used in this study.

**Table 3 Details on the datasets used in this study**

| Data type | Content | Resolution | Span | Data usage | Data sources |
|---|---|---|---|---|---|
| Winter wheat land cover data | Classification of winter wheat | 30m | 2016-2020 | Research area | Dong et al., 2020a |
| Satellite data | $EVI2_{max}$ | 30m | Winter wheat growing season of each year from 2016 to 2021 | Input variables | Landsat 8 and Sentinel 2 dataset of GEE platform |
| Meteorological data | Tem Rad Pre | 0.1° | March to May of each year from 2016 to 2021 | Input variables | ERA5 dataset of GEE platform |
| In-situ measured yield data | Field-level yield with coordinates | Field-level | 2016-2021 | Model establishment and evaluation | Field investigation |
| Census yield data | Statistical data | Province-level and municipal-level | 2016-2021 | Model validation | State statistical bureau |
| Yield dataset | GlobalWheatYield4km | 4km | 2016-2020 | Dataset comparison | Luo et al., 2022 |

190

## 2.3 Method

### 2.3.1 Methodology

The hierarchical linear model (HLM) is a simple and efficient method for dealing with nested structures. At present, HLM has been extensively applied to predicting yield, grain protein content, and agronomic traits for inter-annual and transregional (Li et al., 2020; Xu et al., 2020; Li et al., 2022; Zhao et al., 2022b). These papers have demonstrated that the HLM method is a stable, reliable and scalable way of solving yield estimation problems. They also demonstrated that, although a linear relationship between $EVI2_{max}$ and crop yield can be established in a particular field of a single year, differences in meteorological factors between regions and years will differentiate this relationship, which is the exact problem that the HLM model was implied to settle. In this study, normalization was performed on the data to reduce the impact of differences in variable scales. For each province, a set of parameters was generated by using the data collected from the sample fields. The specific yield-predicting models in different provinces using the HLM method in this study involved a two-levels hierarchy. Level 1 of the HLM model was constructed based on the yield and $EVI2_{max}$:

$$Level\ 1: Yield\ = \beta_0 + \beta_1 * EVI2_{max} + r \tag{2}$$

where $\beta_0$ and r represent the intercept and random error, respectively, and $\beta_1$ represents the slope of the linear model corresponding to $EVI2_{max}$.

In the HLM, the parameters of $\beta_0$ and $\beta_1$ at Level 1 become dependent variables at Level 2. The independent variables of Level 2 are the accumulated meteorological data (Tem, Rad, and Pre) of different growth stages, such that:

$$Level\ 2: \beta_j\ = \gamma_{mj} + \gamma_{mj} * Tem_{mj}\ + \gamma_{mj} * Rad_{mj}\ + \gamma_{mj} * Pre_{mj} + \mu_{mj} \tag{3}$$

where $\beta_j$ represents the $\beta_0$ and $\beta_1$ from Level 1 of HLM, j represents 0 or 1. $\gamma_{m0}$ is the intercept, and $\gamma_{m1}$ - $\gamma_{m3}$ represent slopes of each accumulated meteorological data of different months (m=3, 4, and 5) and $\mu_{mj}$ is the random error of Level 2 of HLM. The parameters of the HLM model in this article are estimated using maximum likelihood estimation. Figure 2 shows a schematic of the workflow.

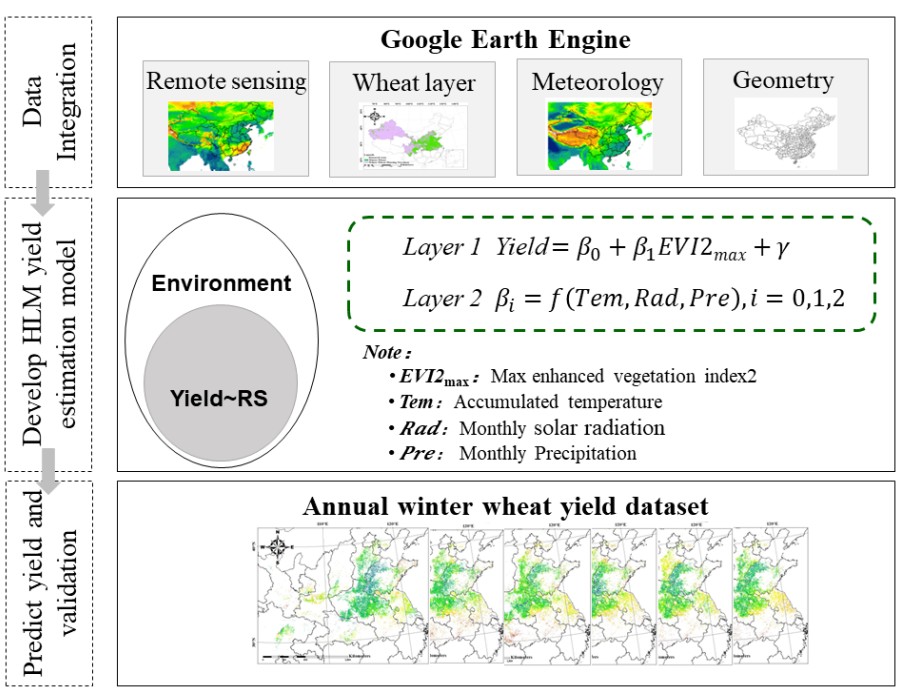

**Figure 2 Schematic diagram outlining the inputs, major processing steps used, and generated outputs.**

### 2.3.2 Comparison with the Random Forest method and the other yield datasets

Random Forest (RF) is a model with predictive performance commonly used in the current yield estimation literature (Li et al., 2020; Cheng et al., 2022; Luo et al., 2022). RF regression is a classic ensemble machine learning model that establishes multiple unrelated decision trees by randomly extracting samples and features and obtains the prediction results in parallel. Each decision tree can obtain a prediction result through the samples and features extracted, and the regression prediction result of the whole forest can be obtained by averaging the results of all trees (Breiman, 2001). This study generated multiple RF models for each province just like the way we build HLM models, using same calibration and validation datasets, so it makes two models for each province and definitely comparable. Given the wide range of RF applications in generating crop yield data, we built a RF prediction model in Matlab and compared its performance with the HLM model. The number of decision trees was set to 200, and the maximum depth of the tree and the number of features were optimized the models' hyperparameters through pretuned procedure (Li et al., 2021; Cheng et al., 2022).

We compared our yield production (ChinaWheatYield30m) with an existing 4-km dataset of global wheat yield (GlobalWheatYield4km) (Luo et al., 2022) using in situ data to validate the reliability of our dataset. More specifically, we calculated the r and rRMSE between the in situ measurement yields and the estimates of GlobalWheatYield4km or ChinaWheatYield30m from 2016 to 2021. This study compared and analysed national statistical data at different scales, focusing mainly on the provincial and municipal levels, to validate the accuracy of the ChinaWheatYield30m dataset. This study compared the difference between statistical yield per unit area from 2016 to and the average yield using ChinaWheatYield30m extracted from both province and municipal vector data. The provincial and municipal average yields based on the ChinaWheatYield30m dataset were calculated by dividing the total yield of all winter wheat pixels by the number of winter wheat pixels in that area.

### 2.3.3 Model evaluation

The commonly used correlation coefficient (r) and relative root mean square error (rRMSE) were used
to compare the performance of generated models. To estimate the contribution of each input variable of
the HLM, we applied an extended Fourier amplitude sensitivity test (Saltelli et al., 1999). The EFAST
(Extended Fourier amplitude sensitivity test) was used to determine a sensitivity index (SI) which
combined the advantages from both Fourier amplitude sensitivity test and Sobol algorithm. The derived
SI quantified how output results were impacted by input variables. The SI of each independent input
variable to the yield in different provinces was computed with Simlab (version 2.2.1) software. To verify
the stability of the yield model in this study, in addition to using independent samples for validation, we
also selected cross-validation of the model deviation in different agricultural regions and years (Fushiki.,
2011). In this study, regional and temporal cross-validation was performed by training the models on
specific years or regions and then independently validating them on the remaining years or study regions
as separate samples.

## 3 Results

### 3.1 Exploring the appropriate method and accuracy assessment

The performance of RF and HLM models in situ yield predictions during 2016 – 2021 for each province
are shown in Fig. 3. The calibration sets for RF and HLM models have similar performance, with r
(rRMSE) ranges of 0.79 - 0.92 (5.78% - 23.37%) and 0.67 - 0.87 (4.87% - 22.06%), respectively.
However, in the validation set, the HLM model outperformed RF with the r (rRMSE) range of 0.50 -
0.93 (1.93%-23.00%) and 0.27 - 0.76 (13.44% - 30.86%), respectively. The superior performance of
HLM was attributed to its ability to capture the interaction effects among various factors. This interaction
explained most of the variation among the provinces, with a sensitive index range of 9.85% - 69.92%
(Fig. 4). The sensitive index of input variables to the HLM model is shown in Fig. 4, indicating the
contributions of each variable to the HLM model. Overall, in most of the analyzed provinces, EVI2 was
the most important variable in the HLM model, with a contribution range of 11.70 % - 63.18% for
different provinces. As for the meteorological factors, in general, temperature was the most important
factor, whereas radiation and precipitation were less significant. The variables related to accumulated
temperature, Tem04 and Tem05, had a high contribution (8.50% - 21.90%) to the HLM model. The
results show the importance of weather in April and May, which in our research areas are the key months
for the flowering and filling of winter wheat, the critical periods in grain formation when most organic
matter is accumulated (Cabas et al., 2010).

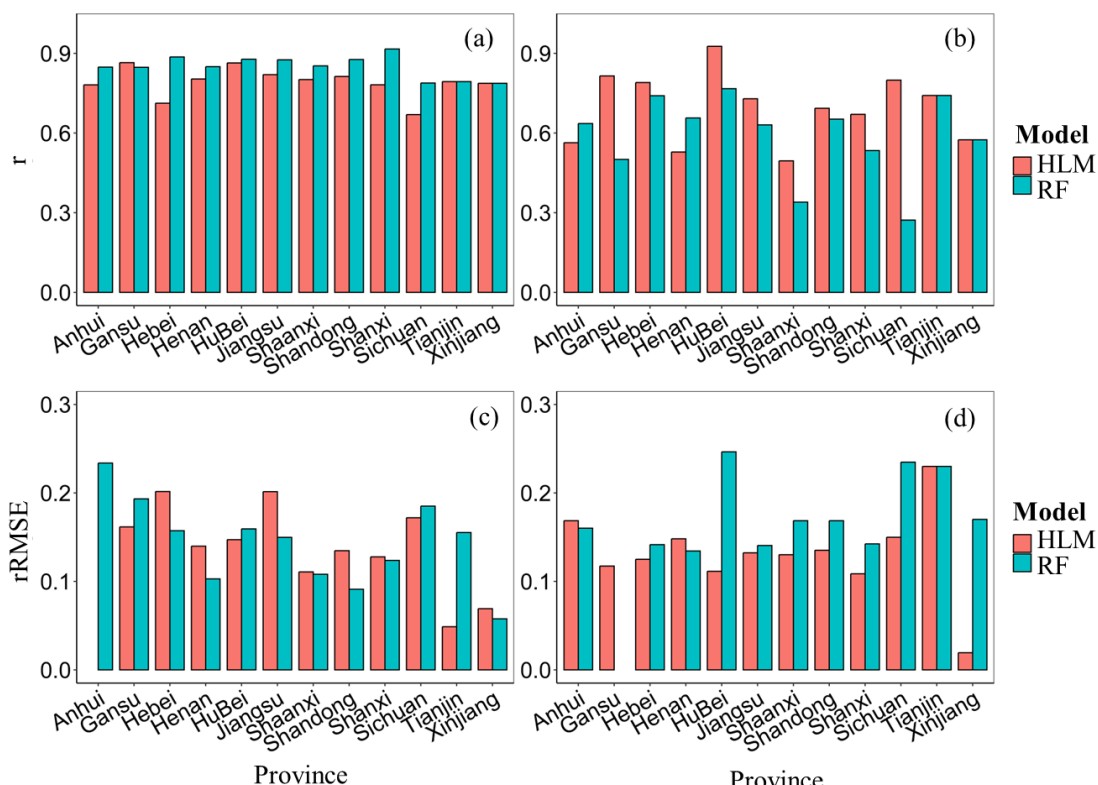

**Figure 3. Comparison between the predicted and measured yield in the calibrated datasets (a, c) and the validation datasets (b, d).**

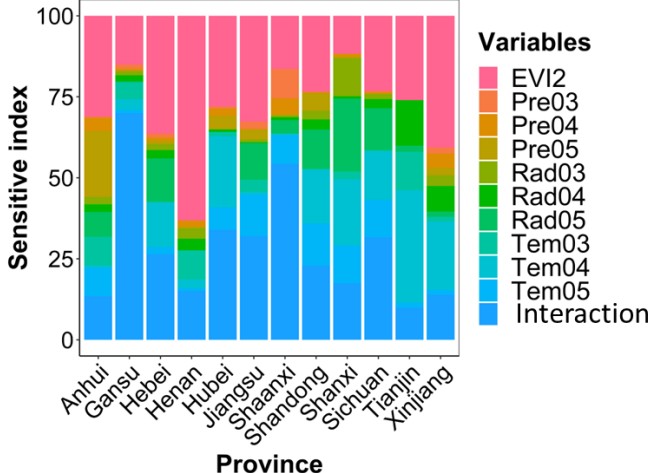

**Figure 4 Sensitive index in the trained HLM model for different input variables.**

The HLM model and RF model was implemented to predict in situ wheat yield using the calibration
dataset. By comparing the predicted results from 2016 to 2021 with the in situ records, it was found that
there is a high consistency between the measured and predicted yield of winter wheat. The r (p<0.01)
and rRMSE for the HLM model were 0.81** and 12.59%, respectively, while for the RF model, the r
(p<0.01) and rRMSE were 0.83** and 12.66%, respectively. When validating with independent samples,
the HLM model performed better than the RF model, with an r (p<0.01) of 0.72** and an RMSE of
15.34% for the HLM model, while the RF model had an r of 0.69** and an RMSE of 15.71%. Due to
the fact that the majority of the pixels to be predicted are located in areas not covered by the calibrated

dataset, the HLM model with stable performance in independent sample validation was chosen for subsequent analysis and dataset construction.

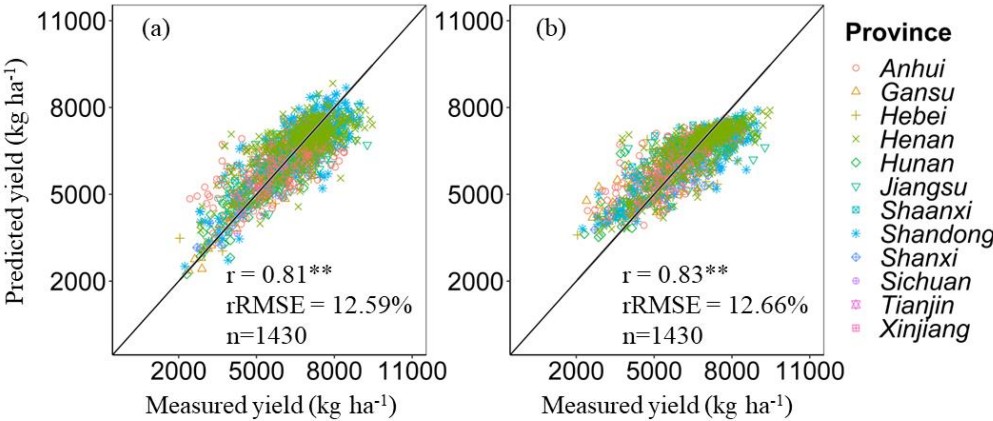

Figure 5. Comparison of measured yield with predicted yield based on HLM model (a) and RF model (b) in the calibrated datasets. ** represents model significant at the 0.01 level of probability.

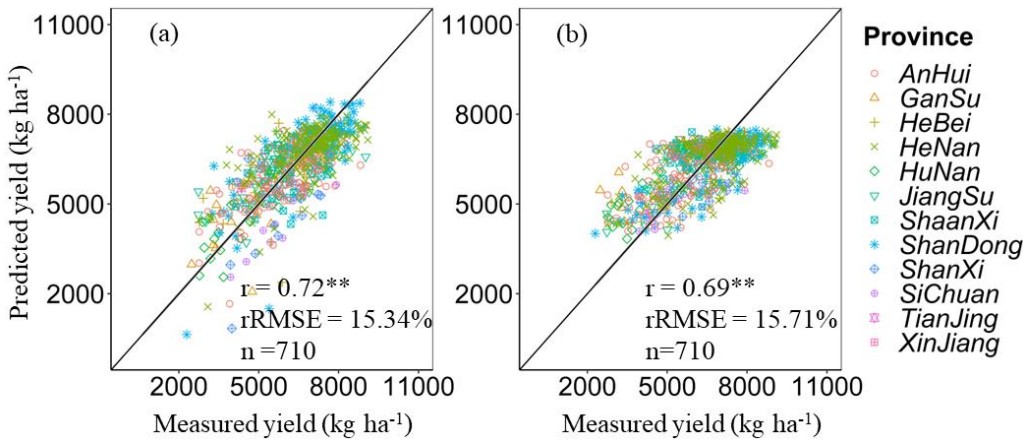

Figure 6. Comparison of measured yield with predicted yield based on HLM model (a) and RF model (b) in the validated datasets. ** represents model significant at the 0.01 level of probability.

**3.2 Cross-validation of the HLM model across years and regions**

Apart from validating the model using independent samples, this study also cross-validated based on different years and different agricultural regions to further assess the stability of the HLM model (Fig. 7 and Fig.8). Interannual cross-validation results show that the predicted yield using the HLM model has high consistency with the measured yield, with r (p < 0.01) and rRMSE values range of 0.55** – 0.69** and 15.44% – 28.61%, respectively. In the regional cross-validation, the cross-validation results in GX regions performed poorly, and the measured data and verification data in other regions have high consistency, with r (p < 0.01) and rRMSE values range of 0.30** – 0.51** and 17.31% – 23.16%, respectively. The yield estimation results for the GX region and the Southwest region are poor. These two regions have a large area, and there are significant differences in climate and planting management conditions. The existing data is not sufficient to reflect these differences. However, the main recommended winter wheat varieties at the provincial level have similar characteristics, and the planting patterns are similar due to policy reasons. By utilizing meteorological conditions, it is possible to reflect

the differences in winter wheat production within provinces as much as possible. Therefore, this article constructed a 30m winter wheat yield dataset for China at the provincial scale.

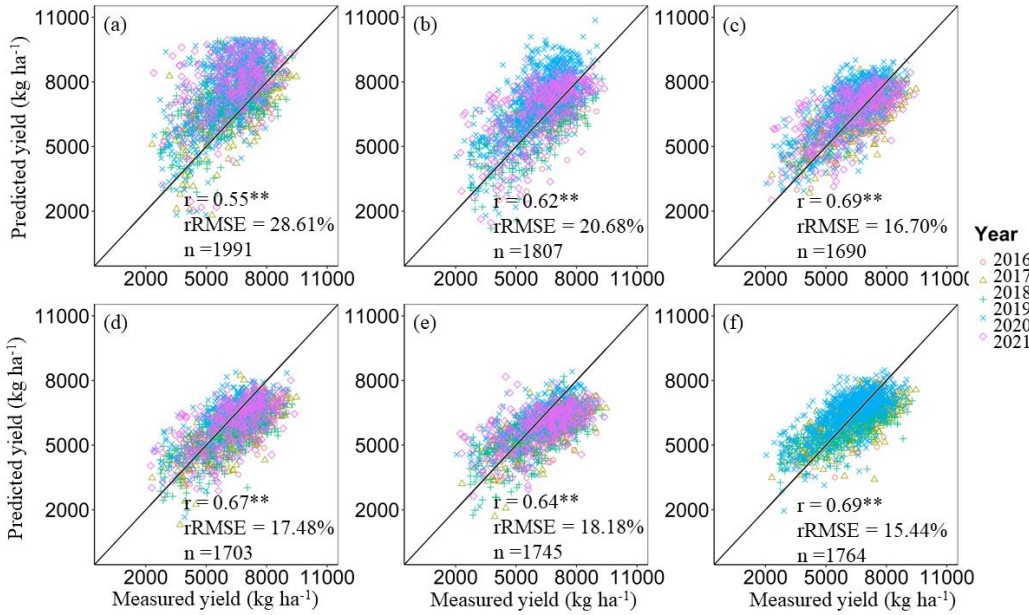

**Figure 7. Interannual cross-validation of the correlation between measured data and predicted data, where (a), (b), (c), (d), (e) and (f) indicate that the HLM models of 2016, 2017, 2018, 2019, 2020 or 2021 are directly used in other years. ** represents model significant at the 0.01 level of probability.**

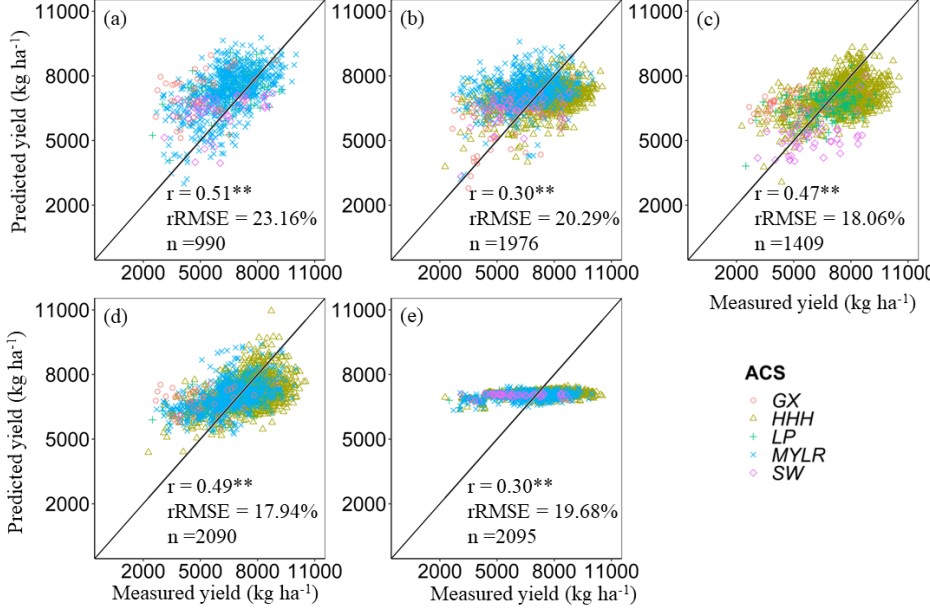

**Figure 8. Reginal cross-validation of the correlation between measured data and predicted data, where (a), (b), (c), (d) and (e) indicate that the HLM models of HHH, LP, MYLR, SW or GX are directly used in other years. ** represents model significant at the 0.01 level of probability.**

## 3.3 Comparing ChinaWheatYield30m with GlobalWheatYield4km

Figure 9 shows the spatial patterns of ChinaWheatYield30m from 2016 to 2021. Generally, the spatial patterns of predicted yields were consistent with in situ measured yields, with large variability from

2273.82 – 10518.82 kg ha$^{-1}$. We further summarized the province-level statistic yield. The yield averages were highest in Shandong Province (6567.48 kg ha-1), followed by Henan Province (6498.42 kg ha$^{-1}$) and Hebei Province (6039.39 kg ha$^{-1}$). By contrast, Jiangsu Province achieved the lowest average yield 325 (4337.05 kg ha$^{-1}$) (Fig. 10). Overall, these data are consistent with the census data. In contrast, model performance showed overestimates of wheat crop yield compared with statistical yield (r = 0.69** (p < 0.01), rRMSE = 19.16%) (Fig. 10). Therefore, the field-scale yield prediction dataset has not only high precision at a fine scale, but also performs well on a large scale.

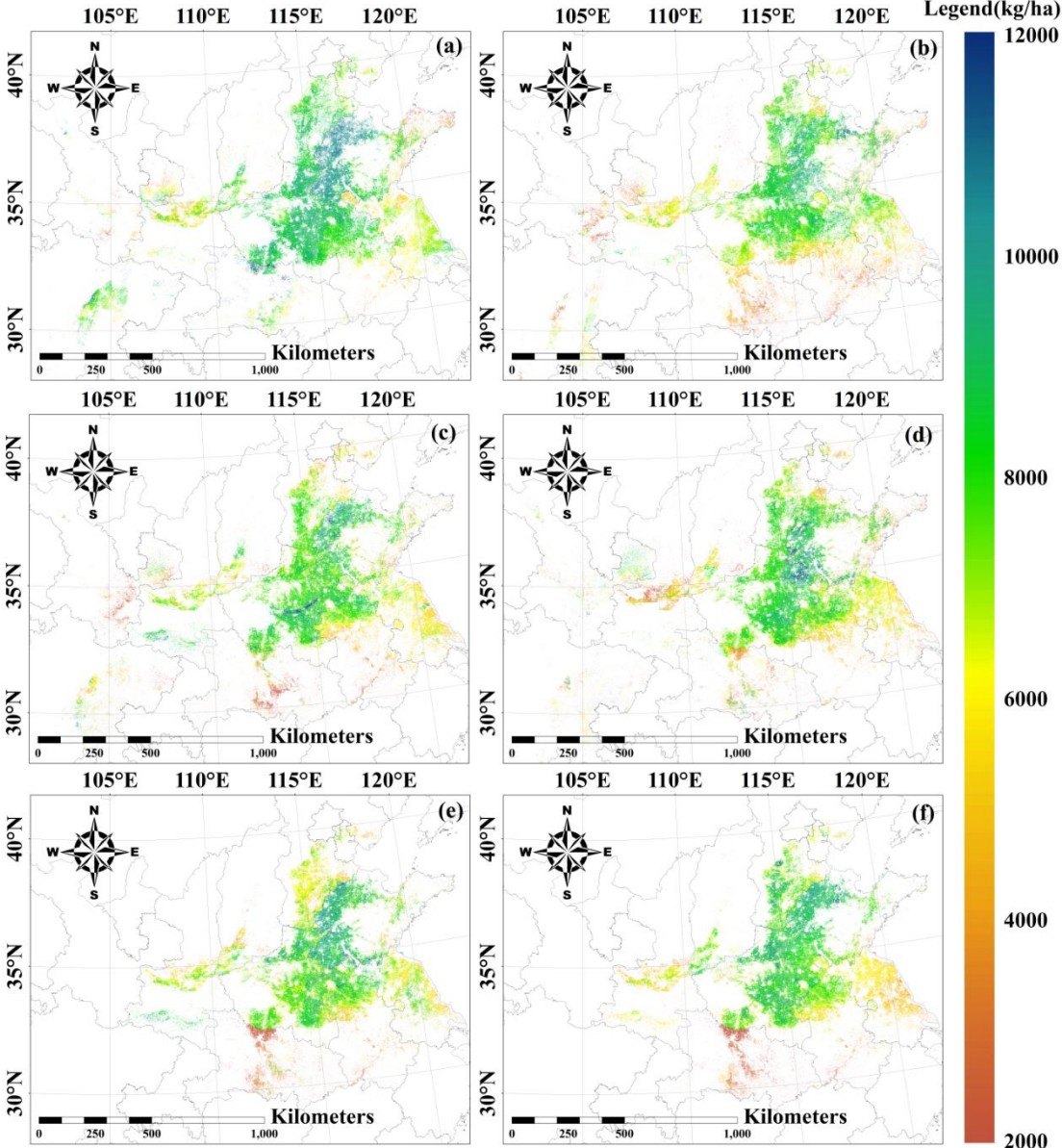

**Figure 9. Spatial patterns of annual winter wheat yield during 2016 - 2021.**

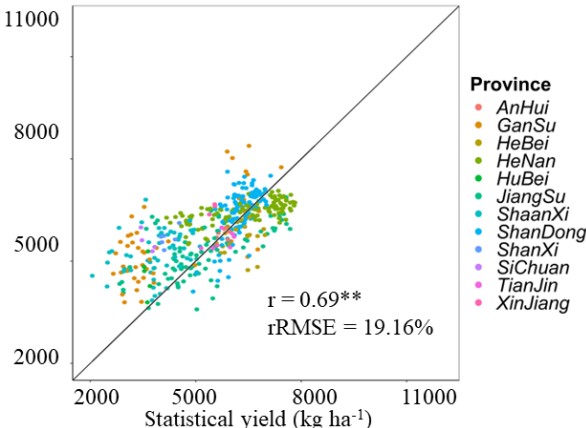

**Figure.10. Comparison of predicted yield and municipal statistical yield. \*\* represents model significant at the 0.01 level of probability.**

We compared the datasets at the field level using single pixels and through a zonal analysis of three
selected research areas. Field-level yield estimates were aggregated to match the ChinaWheatYield30m and GlobalWheatYield4km from 2016 to 2020 and then compared with in situ measurement yields (Fig.11). The yield estimates of ChinaWheatYield30m showed higher consistencies with in situ measurement yields as the scatter points were closer to the 1:1 line than in the case of GlobalWheatYield4km. The results showed that, in different years, ChinaWheatYield30m has a lower
rRMSE range (12.40% – 13.84%) compared to GlobalWheatYield4km (20.43% – 33.06%) (Fig. 9).

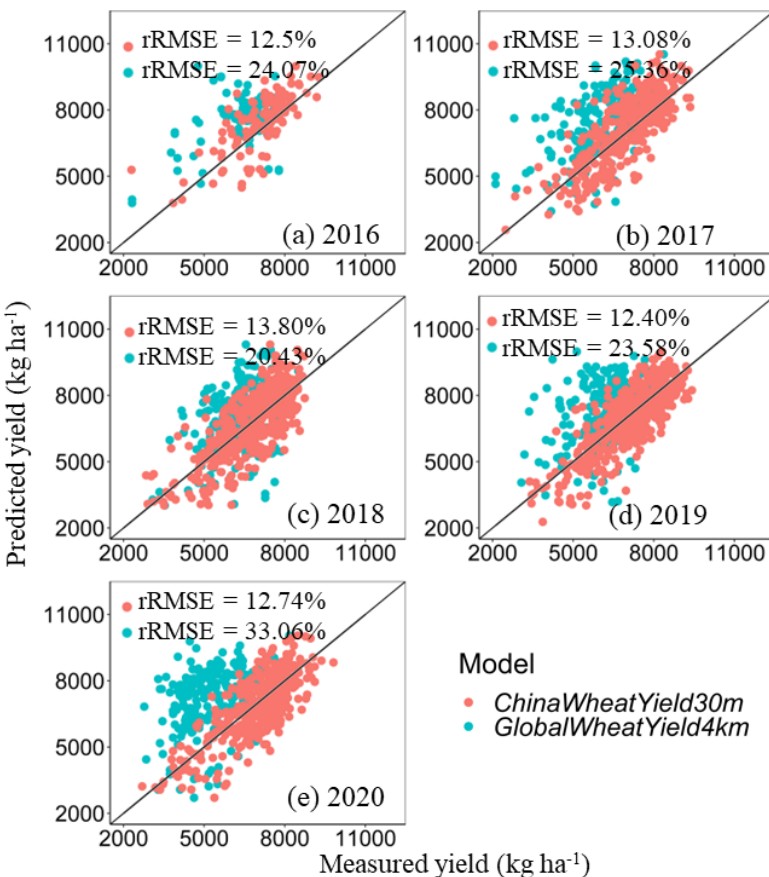

**Figure 11. Comparisons between in situ measurement yields and predicted yields of GlobalWheatYield4km or ChinaWheatYield30m for 2016 (a), 2017 (b), 2018 (c), 2019 (d), and 2020 (e).**

As for the zonal analysis, winter wheat yield derived from ChinaWheatYield30m also have a close spatial pattern to GlobalWheatYield4km production (Fig. 12 and Table 4). Besides, ChinaWheatYield30m, with a standard deviation of 290.27 – 880.91 kg ha$^{-1}$, depicts the difference in yield with greater spatial detail compared to the GlobalWheatYield4km standard deviation of 195.46 – 1516.09 kg ha$^{-1}$. In the selected sample areas, the yield ranges of ChinaWheatYield30m and GlobalWheatYield4km are 2115.95 kg ha$^{-1}$ – 7668.69 kg ha$^{-1}$ and 2653.62 kg ha$^{-1}$ – 10504.50 kg ha$^{-1}$, respectively. This wide range and minor deviation reveal the advantages of fine-resolution data. Compared with the actual yield records, GlobalWheatYield4km significantly underestimates them, whereas ChinaWheatYield30m is closer to the 1:1 line. In the selected sample areas, the mean yield of ChinaWheatYield30m is generally higher than that of GlobalWheatYield4km because the wheat classification at 30-m resolution is dominated by pure wheat pixels. In contrast, the wheat classification with 4-km resolution has more mixed pixels. For example, buildings and roads cannot be identified in the 4-km classification but result in an underestimation of yield prediction (Fig. 12).

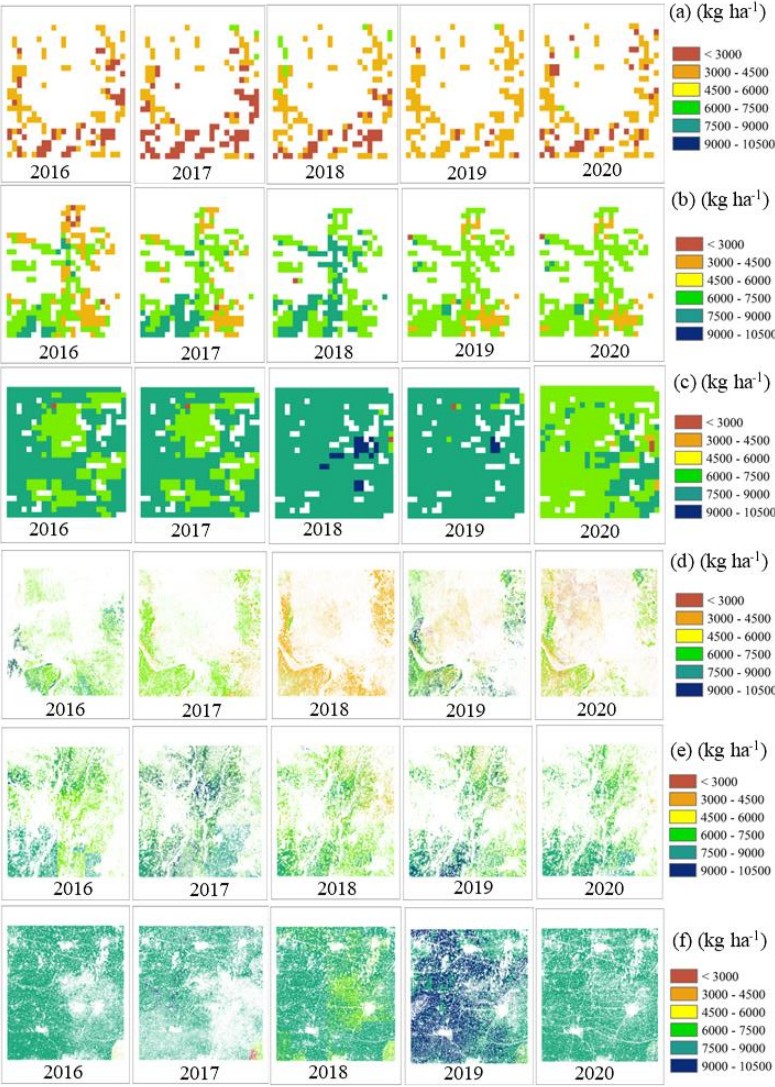

**Figure 12 Comparison of spatial patterns between GlobalWheatYield4km (a, b, c) and ChinaWheatYield30m (d, e, f) from 2016 to 2020. The detailed location of the selected example areas (Region 1 and d; Region 2 and e; Region 3) is shown in Figure 1.**

**Table 4 Statistical analysis of GlobalWheatYield4km and ChinaWheatYield30m**

| Reg. | Year | GlobalWheatYield4km (kg ha[-1]) | | | | ChinaWheatYield30m (kg ha[-1]) | | | |
|---|---|---|---|---|---|---|---|---|---|
| | | Min | Max | Mean | Std | Min | Max | Mean | Std |
| 1 | 2016 | 2215.59 | 4499.56 | 3085.45 | 394.45 | 3787.87 | 10504.50 | 5797.42 | 711.83 |
| | 2017 | 2115.95 | 5543.09 | 3034.08 | 660.67 | 3555.29 | 7470.59 | 4849.84 | 374.35 |
| | 2018 | 2461.41 | 5192.57 | 3499.66 | 632.27 | 3015.54 | 6231.35 | 3746.34 | 422.38 |
| | 2019 | 2802.90 | 4987.77 | 3511.21 | 346.15 | 2653.62 | 9978.73 | 5351.96 | 1516.09 |
| | 2020 | 2336.31 | 4584.65 | 3347.63 | 505.31 | 2705.24 | 7874.20 | 4238.19 | 977.18 |
| 2 | 2016 | 2751.27 | 6626.20 | 4807.49 | 880.91 | 4257.01 | 9078.25 | 6002.05 | 438.40 |
| | 2017 | 3504.07 | 7102.54 | 5349.48 | 847.29 | 4997.04 | 10504.47 | 6564.42 | 968.29 |
| | 2018 | 4524.76 | 6755.62 | 5880.72 | 402.58 | 3818.12 | 10291.08 | 6472.96 | 721.93 |
| | 2019 | 3988.76 | 6555.61 | 5551.69 | 528.77 | 3198.47 | 9902.78 | 6704.211 | 989.46 |
| | 2020 | 3766.66 | 6301.66 | 5069.00 | 526.35 | 4352.21 | 8439.71 | 6100.75 | 745.51 |
| 3 | 2016 | 4388.15 | 7127.87 | 6103.27 | 491.77 | 3788.11 | 7554.13 | 7047.07 | 321.38 |
| | 2017 | 5000.56 | 7387.55 | 6261.93 | 433.99 | 5917.13 | 8266.23 | 7199.44 | 214.30 |
| | 2018 | 5637.92 | 7668.69 | 6931.35 | 356.61 | 4927.40 | 8384.25 | 6357.63 | 378.09 |
| | 2019 | 5589.33 | 7540.64 | 6535.69 | 290.27 | 5394.00 | 9980.07 | 7576.74 | 652.95 |
| | 2020 | 3861.44 | 7003.86 | 5590.34 | 521.12 | 5557.38 | 8186.71 | 6802.47 | 195.46 |

Note: Reg represents Region.

## 4 Discussion

### 4.1 Advancements of the 30-m resolution yield dataset

Information on the spatial extent of winter wheat yield is essential for drafting economic and food subsidy policies and rationally allocating resources (FAOSTAT, 2018). To our knowledge, to date there is no fine resolution (30 m) winter wheat yield distribution map. Previous research has generated the winter wheat yield distribution map of some major production areas in China at moderate resolution, e.g., 10-km, 5-arcmin grid, 5-minute grid, 4-km, and 1 km (Monfreda et al., 2008; Fischer et al., 2012; You et al., 2022; Grogan et al., 2022; Luo et al., 2022; Cheng et al., 2022). Moderate-resolution yield maps have a mixed-pixel problem, which may lead to great uncertainties, as mentioned in comparison with the 4-km yield dataset. Existing wheat yield maps are usually available at the end of the season or based on yield statistics, which limits their application in early field management and government macro-control (Battude et al., 2016; Kang and Ozdogan., 2019). For example, crop growth models strongly depend on daily meteorological data as input; this increases the difficulty in early yield prediction because meteorological data during the season is lacking and long-term meteorological forecasts are unreliable. ChinaWheatYield30m had the following advantages:

1) This study generated ChinaWheatYield30m dataset with 30-m resolution (Fig.10), the primary reason is we adopted winter wheat classification map from (Yuan et al., ESSD 2020), providing highest resolution of 30-m wheat pixels. Such a resolution will provide not only higher result credibility, but also

balance the computational efficiency problems. High-resolution yield datasets can provide more accurate spatial information about crop production, improving agricultural productivity and enabling rapid monitoring and analysis of large agricultural areas. This allows for timely detection and resolution of issues that arise during crop growth, ultimately enhancing both the efficiency and effectiveness of agricultural production.

2) A stable accuracy at field scale and large regional scale will highly contributing to field management, modelling agricultural systems, drafting agricultural policies. This study combined remote sensing and meteorological data to construct a spatiotemporally expandable HLM method for predicting winter wheat yield in the main producing areas. The relationship between vegetation index and crop yield varies across different years and regions (Li et al., 2020). Meteorological data has an important impact on crop yield (Moschini and Hennessy, 2001; Lee et al., 2013). Li et al. (2021) showed that environmental data for wheat in China explained more than 60% of the variation in wheat yield. In this study, we generated ChinaWheatYield30m with stable results, which fully exploited the advantages of HLM to solve the nested problem of yield prediction impacted by remote sensing and meteorological data.

3) The product has a high real-time performance and can be used to forecast the output in the early period of the year. $EVI2_{max}$ and meteorological data used in this paper can be obtained before May, while wheat in China's main winter wheat production areas is generally harvested in June. Therefore, the proposed method can accurately predict winter wheat yield in real time. The strengths of the HLM model are overcoming inter-annual and regional variations (Li et al., 2020; Xu et al., 2021; Zhao et al., 2022b). The results based on field investigation and statistical data show that the method can accurately predict winter wheat yield in the main production areas. The ChinaWheatYield30m is presumed to be most commonly concerned in metropolis level or county level, in this sense, the resolution will be feasible to these scales.

## 4.2 Uncertainties and limitations

Despite the advantages of ChinaWheatYield30m, the dataset also presents some data and model uncertainties.

1) Remote sensing and meteorological data used in this study still have uncertainties. This study generated ChinaWheatYield30m dataset with 30-m resolution, the primary reason is we adopted winter wheat classification map from (Yuan et al., ESSD 2020), providing highest resolution of 30-m wheat pixels. The ChinaWheatYield30m input data consist of meteorological variables and remote sensing data, all datasets were resampled to a 30-m resolution to ensure data uniformity.  In terms of remote sensing data, resampling Sentinel 2 data to 30 meters may result in loss of some surface information, and the differences between pixels in the image may not be accurately captured. The increase in the number of mixed pixels can lead to uncertainties in yield estimation results. Besides, maximum EVI2 is obtained at the heading or flowering period (Luo et al., 2020), but due to the irregular availability of usable Sentinel 2 and Landsat 8 observations, the maximum EVI2 nationwide may correspond to different phenological periods.In addition, meteorological data is another important component of the yield dataset. To obtain spatially and temporally continuous meteorological driving data, this study utilizes a dataset generated by ECMWF, its meteorological data was timely updated to meet our spatio-temporal demand. However, meteorological data such as precipitation, temperature, and radiation exhibit highly nonlinear and chaotic

characteristics (Lorenz, 1996), leading to ongoing debates about the reliability of interpolation methods. The coarse resolution of meteorological data, combined with its high spatial homogeneity over larger areas, weakens its ability to effectively capture the relationship between remote sensing data and yield variations as the second-level correction in the HLM model.

2)Uncertainties in winter wheat classifications are transferred to the yield predictions. The wheat classification is based on optical remote sensing data and may be affected by meteorological factors such as clouds and rain (Dong et al., 2020a). In addition, the winter wheat classification data are mainly based on time series, and a similar time series may lead to a wrong classification, which results in uncertainties in regional yield statistics.

3) The accessibility of in situ measurement data is also one of the uncertainties in ChinaWheatYield30m. The performance of HLM depends on the quantity and quality of samples. It is more precise when sampling in the quadrat and is often higher than the statistical yield data. It was particularly difficult to collect finer-scale census data with longer time coverage in some areas, such as Xinjiang Province, leading to data gaps in ChinaWheatYield30m. We combined in situ measurements and statistical data to

calibrate and validate the ChinaWheatYield30m. However, where sparse observation where available, we could only calibrate the parameters of the mathematical optimization.

4) The uncertainties of HLM application scenarios need further analysis. There is a nested issue between vegetation indices and yield relationships, as well as between meteorological data and yield relationships (Li et al., 2020; Xu et al., 2020). HLM has advantages in addressing this problem. Under similar

meteorological conditions, the yield estimation of the model mainly depends on the differences in vegetation indices. In the major wheat production area, variations in crop types, soil types, climate factors, and other factors have an impact on the model's estimation results (Li et al., 2021). The current model only considers the effect of meteorological data on remote sensing yield estimation, and future analyses will incorporate additional factors such as soil to generate more accurate yield datasets. The current

model is primarily constructed based on normal production conditions, and estimating winter wheat yield under abnormal climatic conditions introduces significant uncertainties. Therefore, it is necessary to consider stress factors and further improve the framework of remote sensing estimation models for winter wheat in the future.

**5 Data availability**

The derived yield dataset for ChinaWheatYield30m during 2016 – 2021 is available at https://doi.org/10.5281/zenodo.7360753 (Zhao et al., 2022a). Please be so kind to contact the authors for more detailed information.

**6 Conclusions**

In the present study, we generated a 30m Chinese winter wheat yield from 2016 to 2021 based on the

HLM model, called ChinaWheatYield30m. First, we construct a semi-mechanical model with excellent accuracy and low cost in a combination of RS observations and regional meteorological information for major winter wheat-producing areas in China. The HLM model has stable performance in calibration

sets across China, with r of 0.81** (p < 0.01) and rRMSE of 12.59%, respectively. Next, we validated the predictive performance of in-situ measurement data and statistical data. The ChinaWheatYield30m dataset was highly consistent with in-situ measurement data and statistical data (p < 0.01), indicated by r (rRMSE) of 0.72** (15.34%) and 0.69** (19.16%), respectively. Finally, we established a high-resolution yield product for winter wheat in China during 2016 – 2021. Our ChinaWheatYield30m can be applied for many purposes, including further academic research, making economic, food subsidy policies and rationally allocating imperative resources.

**Author contributions**

YZ, GY and SH designed the research, performed the analysis, and wrote the paper; JZ, HX, YM and XL collected datasets; GY, XY, XX, ZL and SC performed the analysis; GY edited and revised the paper.

**Competing interests**

The authors declare that they have no known conflict of interest.

**Acknowledgements**

We would like to thank the editor and the two reviewers for their valuable comments. We would also like to thank all the scientists and students who participated in the field observations.

**Financial support**

This research has been supported by National Key Research and Development Program of China (2022YFD2001103), Key scientific and technological projects of Heilongjiang province (2021ZXJ05A05), the Platform Construction Funded Program of Beijing Academy of Agriculture and Forestry Sciences (PT2022-24) and Chongqing Technology Innovation and Application Development Special Project (cstc2019jscx-gksbX0092, cstc2021jscx-gksbX0064.

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
