# Peer review of "ChinaWheatYield30m: A 30-m annual winter wheat yield dataset from 2016 to 2021 in China"

_Earth System Science Data, 2022_

## Author Comment (AC1)

The study aims to generate a 30-m Chinese winter wheat yield dataset (ChinaWheatYield30m) using hierarchical linear model with various input datasets. Results show that the ChinaWheatYield30m dataset was consistent with in-situ measurement data and statistical data, indicated by r (nRMSE) of 0.72** (15.34%) and 0.73** (19.41%), respectively. Overall, there are some aspects needed to be further improved, and the comments are listed below.

1.Line 161-168. How many field-scale yields per year? Where are they distributed? A map is needed for this. And are these fields in the same location each year? If not, a table can be listed to show the number of fields for each year and province.

[Response]: Thank you very much for your suggestion. The sample points may be in the same location each year, or the experimental sampling locations may be increased or modified. In order to more accurately display the sampling points and data in this article, Figure 1 in article has been modified and sample points have been added to the figure. In addition, a table has been added to record the sample numbers of the fields in different years and provinces. We therefore rewrote the following paragraph from Lines 126-128 and table 2:

**Table 2 Detailed statistics on the sample numbers in this study.**

| Province | Anhui | Gansu | Hebei | Henan | Hubei | Jiangsu | Shaanxi | Shandong | Shanxi | Sichuan | Tianji | Xinjiang | Total |
|---|---|---|---|---|---|---|---|---|---|---|---|---|---|
| 2016 | 12 | 8 | 26 | 45 | | 33 | | 10 | 3 | 11 | 1 | | 149 |
| 2017 | 53 | 4 | 35 | 72 | 16 | 46 | 25 | 59 | 11 | 9 | 1 | 2 | 333 |
| 2018 | 85 | 3 | 63 | 126 | 18 | 47 | 21 | 56 | 14 | 13 | 1 | 3 | 450 |
| 2019 | 85 | 3 | 48 | 130 | 13 | 53 | 17 | 62 | 14 | 10 | | 2 | 437 |
| 2020 | 82 | 10 | 26 | 121 | 11 | 60 | 19 | 52 | 14 | 0 | | | 395 |
| 2021 | 81 | 7 | 25 | 125 | 10 | 26 | 18 | 64 | 8 | 7 | 2 | 3 | 376 |
| Total | 398 | 35 | 223 | 619 | 68 | 265 | 100 | 303 | 64 | 50 | 5 | 10 | 2140 |

2.Line 185-186. Why train the model separately for each province, considering that some provinces have a large geographical span and thus may have internal heterogeneity, I think it may be more reasonable to divide them according to agricultural cultivation subdivisions.

[Response]: Thank you very much for your suggestion. China's agricultural regions refer to the division of China into different areas based on factors such as climate, land use, and crop types, in order to scientifically and reasonably organize and develop agricultural production. The main production areas of winter wheat in China are mainly distributed in some areas of the North China region, the Huang-Huai-Hai Plain, the Yangtze River Middle and Lower Reaches region, and the southwestern region. Each main production area of winter wheat includes multiple provinces. There are significant differences in main crop varieties, crop growth and development, and management practices in these regions. The distribution of China's agricultural regions and provinces can be seen in Figure S1. Figure S1(a) and (b) showed that one agricultural region spanning multiple provinces, for example, the Huang-Huai-Hai Plain includes Henan, Hebei and Shandong. These provinces are a smaller unit of one agricultural region, in other words, each province located in the same agricultural region. Therefore, this article trains the yield model at the province scale to maximize the accuracy of yield prediction results.

Of course, we also agree with your consideration of using agricultural regions for yield prediction. Therefore, we used different agricultural regions as standards for cross-validation of yield results, as shown in Figure 7, Line 236-239 and line 281-293. In addition, we also added a discussion on this aspect.

[Figure]

Fig.S1 Map of China's Agricultural Regions (a) and Provinces (b) Distribution.

3.Line 218. Please check, is "relative root mean square error" shorted as "nRMSE" instead of "rRMSE"?

[Response]: Thank you very much for your suggestion. we replaced rRMSE, with nRMSE. The revised sections include all relevant information in the text, figures, and tables throughout the article.

4.Section 3.1 and 3.2, the field-level yield dataset can be further divided into different sets, and then a cross-validation result will better indicate the reliability of the model.

[Response]: Thank you very much for your suggestion. Based on your opinion, in addition to using independent samples for validation, we also selected cross-validation of the model deviation in different agricultural regions. In this paper, commonly used 5-fold cross-validation is used in this study. Modification is incorporated in Line 236-239, Line 281-293 and Figure 7.

5.The province-level validation for this dataset seems to be less meaningful because they are too different in scale and there are too many uncertainties, e.g., crop classification. County or municipal level validation may be more valuable.

[Response]: Thank you very much for your suggestion. The modification has been made according to your opinion, mainly including the acquisition of statistical data, the purpose of comparison, and the comparison results. Modification is incorporated in Line 175-179 and Figure 6.

6.Section 3 is results and discussion, and Section 4 is discussion. I think the title is inappropriate.

[Response]: Thank you very much for your suggestion. We apologized for the mistake. We have changed the title of Section 3 to "Results" and updated and revised the content of the article accordingly.

7.It would be better to present the exact location of the in situ measurement.

[Response]: Thank you very much for your suggestion. We fully understand the purpose of presenting the in situ measurement points, so we have added Table 2 and modified Figure 1 to better display the spatial distribution of the data and sample number.

8.Discussion section is insufficient, especially the uncertainty analysis. For example, how do author deal with these datasets with different spatial resolution, does this bring uncertainties to the final results? Is there any deficiency in HLM model? Etc.

[Response]: Thank you very much for your suggestion. To further discuss the potential uncertainties in the dataset input variables, model, and results, the article analyzed various aspects including dataset resolution, classification dataset, survey samples, and model structure. Modification is incorporated in Line 378-418, as follows:

"*1) Remote sensing and meteorological data used in this study still have uncertainties. This study generated ChinaWheatYield30m dataset with 30-m resolution, the primary reason is we adopted winter wheat classification map from (Yuan et al., ESSD 2020), providing highest resolution of 30-m wheat pixels. The ChinaWheatYield30m input data consist of meteorological variables and remote sensing data, all datasets were resampled to a 30-m resolution to ensure data uniformity. In terms of remote sensing data, resampling Sentinel 2 data to 30 meters may result in loss of some surface information, and the differences between pixels in the image may not be accurately captured. The increase in the number of mixed pixels can lead to uncertainties in yield estimation results. Besides, maximum EVI2 is obtained at the heading or flowering period (Luo et al., 2020), but due to the irregular availability of usable Sentinel 2 and Landsat 8 observations, the maximum EVI2 nationwide may correspond to different phenological periods.In addition, meteorological data is another important component of the yield dataset. To obtain spatially and temporally continuous meteorological driving data, this study utilizes a dataset generated by ECMWF, its meteorological data was timely updated to meet our spatio-temporal demand. However, meteorological data such as precipitation, temperature, and radiation exhibit highly nonlinear and chaotic characteristics (Lorenz, 1993), leading to ongoing debates about the reliability of interpolation methods. The coarse resolution of meteorological data, combined with its high spatial homogeneity over larger areas, weakens its ability to effectively capture the relationship between remote sensing data and yield variations as the second-level correction in the HLM model.*

*2)Uncertainties in winter wheat classifications are transferred to the yield predictions. The wheat classification is based on optical remote sensing data and may be affected by meteorological factors such as clouds and rain (Dong et al., 2020).*

*…*

*4) The uncertainties of HLM application scenarios need further analysis. There is a nested issue between vegetation indices and yield relationships, as well as between meteorological data and yield relationships (Li et al., 2020; Xu et al., 2020). HLM has advantages in addressing this problem. Under similar meteorological conditions, the yield estimation of the model mainly depends on the differences in vegetation indices. In the major wheat production area, variations in crop types, soil types, climate factors, and other factors have an impact on the model's estimation results (Li et al., 2021). The current model only considers the effect of meteorological data on remote sensing yield estimation, and future analyses will incorporate additional factors such as soil to generate more accurate yield datasets. The current model is primarily constructed based on normal production conditions, and estimating winter wheat yield under abnormal climatic conditions introduces significant uncertainties. Therefore, it is necessary to consider stress factors and further improve the framework of remote sensing estimation models for winter wheat in the future.*"

9.Some minor formatting issues: Line 163, "1 m$^2$ per point", Line 197-198, "$\beta mj$", "$\gamma m0$",…, some subscripts are not displayed correctly.

[Response]: Thank you very much for your suggestion. Modification is incorporated in manuscript.

---

## Author Comment (AC2)

This article generated a 30m Chinese winter wheat yield from 2016 to 2021 based on the HLM model, called ChinaWheatYield30m. The semi-mechanical model was constructed in a combination of RS observations and regional meteorological data for major wheat-producing regions in China. The ChinaWheatYield30m dataset is validated and has a potential to be applied in some related academic researches.

The paper was basically well organized and written. However, to further improve the paper, some issues need to be deal with. Below are some specific comments:

1.The detailed description is needed to address how to compare the ChinaWheatYield30m dataset with the province-level statistical data;

[Response]: Thank you very much for your suggestion. The primary purpose of statistical data is to verify the accuracy of data sets when statistics are performed at different scales, in order to better serve different institutions for use. Modification is incorporated in Line 223-225. as following:

"This study compared and analyzed national statistical data at different scales, focusing mainly on the provincial and municipal levels, to validate the accuracy of the ChinaWheatYield30m dataset. This study compared the difference between statistical yield per unit area from 2016 to and the average yield using ChinaWheatYield30m extracted from both province and municipal vector data.".

2.The strength of the ChinaWheatYield30m dataset needs to be emphasized comparing with some existed remote sensing yield estimation datasets;

[Response]: Thank you very much for your suggestion. According to your opinion, the article further elaborates on the advantages of the ChinaWheatYield30m. Modification is incorporated in Line 344-375. as following:

"…

1) This study generated ChinaWheatYield30m dataset with 30-m resolution (Fig.10), the primary reason is we adopted winter wheat classification map from (Yuan et al., ESSD 2020), providing highest resolution of 30-m wheat pixels. Such a resolution will provide not only higher result credibility, but also balance the computational efficiency problems. High-resolution yield datasets can provide more accurate spatial information about crop production, improving agricultural productivity and enabling

*rapid monitoring and analysis of large agricultural areas. This allows for timely detection and resolution of issues that arise during crop growth, ultimately enhancing both the efficiency and effectiveness of agricultural production.*

*2) A stable accuracy at field scale and large regional scale will highly contributing to field management, modelling agricultural systems, drafting agricultural policies. This study combined remote sensing and meteorological data to construct a spatiotemporally expandable HLM method for predicting winter wheat yield in the main producing areas. The relationship between vegetation index and crop yield varies across different years and regions (Li et al., 2020). Meteorological data has an important impact on crop yield (Moschini and Hennessy, 2001; Lee et al., 2013). Li et al. (2021) showed that environmental data for wheat in China explained more than 60% of the variation in wheat yield. In this study, we generated ChinaWheatYield30m with stable results, which fully exploited the advantages of HLM to solve the nested problem of yield prediction impacted by remote sensing and meteorological data.*

*3) The product has a high real-time performance and can be used to forecast the output in the early period of the year. EVI2max and meteorological data used in this paper can be obtained before May, while wheat in China's main winter wheat production areas is generally harvested in June. Therefore, the proposed method can accurately predict winter wheat yield in real time. The strengths of the HLM model are overcoming inter-annual and regional variations (Li et al., 2020; Xu et al., 2021; Zhao et al., 2022). The results based on field investigation and statistical data show that the method can accurately predict winter wheat yield in the main production areas. The ChinaWheatYield30m is presumed to be most commonly concerned in metropolis level or county level, in this sense, the resolution will be feasible to these scales."*

3.The Table 1 needs to be reformatted;

[Response]: Thank you very much for your suggestion. Table 1 is now reformatted.

4.A line needs to be inserted between the Table1 and the below paragraph;

[Response]: Thank you very much for your suggestion. A row has been inserted below Table 1 in the article.

5.Line135:"2.1 The winter wheat land cover data" should be "2.2.1";

[Response]: Thank you very much for your suggestion. Based on your feedback, 2.1 has been changed to 2.2.1. Modification is incorporated in Line 138.

6. There is something wrong with the format of the 2.3.1 section, needs to be adjusted.

[Response]: Thank you very much for your suggestion. The format of section 2.3.1 has been modified and all format throughout the entire document has been checked. Modification is incorporated in manuscript.

---

## Author Response (AR2)

Dear editor,

Thank you for your comments concerning our manuscript (MS). We have substantially revised our manuscript with the comments provided by the reviewer. We have studied comments carefully and have made corrections, which we hope meet with approval. Revised portions are marked in red on the paper. The leading corrections in the paper and the response to the reviewer's comments are as follows:

Generating spatial crop yield information is of great significance for academic research and guiding agricultural policy. Here Zhao et al., generated a 30-m winter wheat yield dataset covering main winter wheat-growing region of China from 2016 to 2021. The authors said they proposed a a semi-mechanistic model named HLM that showed better performance against one commonly used machine learning named random forest (RF). With the model developed, and observational dataset including meteorological variables, vegetation index, and yield, they generated the grid-level yield dataset across multiple years. I think this work is important and I would recommend it for publication if my following major concerns, mainly related to model configuration, evaluation, and comparison, could be resolved. The detailed comments are listed as bellows.

Major comments

1) Clarify the model used: section 2.3.1, Eq. 2, what does j represent? Did you develop each HLM separately for each province or city over each month? In Eq.3, what does $\beta mj$ represent in Eq. 2? Did you normalize each input variable before inputting those variables to the model? How did you solve the parameters in Eq.2-3? Please clarify those details and code to make those results reproducible.

[Response]: Thank you very much for your suggestion. The hierarchical linear model (HLM) is a simple and efficient method for dealing with nested structures. In this study, normalization was performed on the data before modeling to reduce the impact of differences in variable scales. For each province, a set of parameters was generated by using the data collected from the sample fields. The specific yield-predicting models in different provinces using the HLM method in this study involved a two-levels hierarchy. βj represents the β0 and β1 from Level 1 of HLM, j represents 0 or 1. The parameters of the HLM model in this article are estimated using maximum likelihood estimation. Modification is incorporated in Line 201-206.

2) Baseline machine learning model configuration: Line 216-219: did you build one RF model for the entire study area and compare it with multiple HLM models? If so, it should not be a fair comparison.

What if you built multiple RF models? In addition, how did you select the key parameters used in RF model, including maximum depth of the tree, the number of features, minimum number of samples required to split an internal node, and minimum number of samples required to be at a leaf node. Note that inappropriate model configuration would generate a worse performance for the RF model. Please clarify how did you select those parameters in details.

[Response]: Thank you for asking, we generated multiple RF models for each province just like the way we build HLM models, using same calibration and validation datasets, so it makes two models for each province and definitely comparable. To clearly demonstrate our approach, the corresponding L224-232. Concerning the key RF parameters, we optimized the models' hyperparameters through pretuned procedure, using 10-fold cross-validation. Majorly we adjust the number of trees and was tuned to 200 trees. There are several relevant studies have indicated the similar results(Li et al., 2021; Cheng et al., 2022). In the data analysis of this study, it has been found that the RF model achieves stable accuracy and small errors with a number of trees below 200.

*Cheng, M., Jiao, X., Shi, L., Penuelas, J., Kumar, L., Nie, C., Wu, T., Liu, K., Wu, W., and Jin, X.: High-resolution crop yield and water productivity dataset generated using random forest and remote sensing, Scientific Data, 9, 641, 10.1038/s41597-022-01761-0, 2022.*

*Li, L., Wang, B., Feng, P., Wang, H., He, Q., Wang, Y., Li Liu, D., Li, Y., He, J., and Feng, H.: Crop yield forecasting and associated optimum lead time analysis based on multi-source environmental data across China, Agricultural and Forest Meteorology, 308, 108558, 2021.*

3) Model validation: since the authors said that the proposed models can overcome inter-annual and cross-regional problems. It's therefore required to validate the model performance by cross-region and cross-year. Line 235-237, did you train the model over part of the studied regions and then validate model performance over the remaining part of the studied regions where the model has never seen the observed yield values (i.e., cross-regional validation to show model overcome the cross-regional problem)? In addition, experiments are required to validate the model performance by training the model in specific years and then validating the model performance in other years that the model has never seen its observed yield data (i.e., cross-year validation to show model overcome the inter-annual problem).

[Response]: Thank you very much for your suggestion. The cross-validation results are based on modeling in a certain year and verification in other years. Region cross-validation models one region and validates other regions. In this study, regional and temporal cross-validation was performed by training

the models on specific years or regions and then independently validating them on the remaining years or study regions as separate samples. Modification is incorporated in Line 253-255, Line 296-298, Fig.7 and Fig.8.

4) Fig.5-6, did you include all the training and validation dataset in this scattered plot or only used the validation dataset for this evaluation? Please clarify it in the main text or in the figure caption. Note that the model performance should be validated against independent validation dataset that the model has never seen. In Fig. 5, please clarify what does each point represent.

[Response]: Thank you very much for your suggestion. In this paper, the data were randomly split into two dataset, two-thirds of the data were used for modelling, and the remaining data were used for validation. This article uses independent samples for model validation, which means that the data used for validation are not included in the modeling data. The titles of Figure 5 and Figure 6 have been updated. Figure 5 presents the results based on the modeling dataset samples, while Figure 6 shows the results based on independent validation data. Additionally, the different colored dots in Figure 5 have also been explained.

5) Fig.5-Fig.7, why not compare RF and HLM using the validation schemes in Fig.5-7? If no such kind of comprehensive comparison, it could be less convincing to draw the conclusion that the proposed method had excellent accuracy.

[Response]: Thank you very much for your suggestion. For the reliability of model comparison, RF and HLM were compared using the same modeling dataset and validation dataset. The results showed that RF had higher accuracy on the modeling dataset, but its stability on the validation results was poor. Therefore, HLM was used for subsequent result presentation and generation of yield datasets. In order to reduce reader misunderstanding, we have reorganized the data analysis results. Modification is incorporated in Fig.3, Fig.5, Fig.6 and Line 281-289.

6) In addition to '**', please explicitly show the p-value for all related results.

[Response]: Thank you very much for your suggestion. In this article, ** represents model significance at the 0.01probability level (p < 0.01). The relevant figures, tables, and descriptions in the article have all been renumbered.

Specific comments

1) Line 19, "a coarse spatial resolution" ranging from what? Ranging from 100 meters to 1 km? Since

the major uniqueness of the dataset is its high spatial resolution, I suggested to show the spatial resolution of previous related dataset in the abstract with few words.

[Response]: Thank you very much for your suggestion. Indeed, showcasing the resolution of existing yield datasets helps highlight the high spatial resolution of this study. Based on your feedback, we have incorporated relevant information into the abstract. Modification is incorporated in Line 19-20.

2) Line 19-21, the sentence is less rigorous. I would revise it as "useful for analyzing large-scale temporal and spatial changes in yield, …deal with small-scale spatial heterogeneity, which …of the Chinese farmers' economy"

[Response]: In accordance with your suggestions, the sentence can be modified as follows: "Although these datasets are useful for analyzing large-scale temporal and spatial change in yield and they cannot deal with small-scale spatial heterogeneity, which happens to be the most significant characteristic of the Chinese farmers' economy." Thank you very much for your suggestion. Modification is incorporated in Line 19-22.

3) Line 29-31, I understand that '**' could be related to significance test (i.e., p-value), but I suggest to remove those symbols in the abstract since you even did not explain it. Note that the same symbol with no explanation could represent different meanings which could confuse the readers.

[Response]: Thank you very much for your suggestion. Based on your feedback, all the "**" symbols have been removed as suggested.

4) Line 68-69, "ML methods need large multidimensional datasets, which can challenge their application", that's not the limitation of ML methods. Note that the inputs of ML models can be the same to parametric regression models or the process-based models. Additionally, there have been many strategies developed for dealing with multi or high-dimensional inputs in ML.

[Response]: Thank you very much for your suggestion. Yield estimation requires a large amount of data, but data generation algorithms have already been developed in ML methods. Based on your feedback, the sentence can be modified as follows: "Overall, ML methods heavily rely on large training datasets (Cao et al., 2021). Nonetheless, the application of machine learning in the realm of synthetic data generation has also exhibited encouraging outcomes (Arslan et al., 2019; Sivakumar et al., 2022; Ebrahimy et al., 2023)". Modification is incorporated in Line 69-71.

5) Line 91, revise "migration learning" as "transfer learning" that is commonly named as

[Response]: Thank you very much for your suggestion. We have revised "migration learning" as "transfer learning". Modification is incorporated in Line 93.

6) Line 99, revise "county to city scale" as "county or city scale" since a county can be larger than a city in US.

[Response]: Thank you very much for your suggestion. We have revised "county to city scale" as "county or city scale". Modification is incorporated in Line 101.

7) Line 109-111, please see my comment#2

[Response]: In accordance with your suggestions, the sentence can be modified as follows: "Although these datasets are useful for analyzing large-scale temporal and spatial change in yield and they cannot deal with small-scale spatial heterogeneity, which happens to be the most significant characteristic of the Chinese farmers' economy." Thank you very much for your suggestion. Modification is incorporated in Line 111-113.

8) Fig. 1, remove the explanation of "sampling points" in the title of the figure since it has been explained in the figure legend. In addition, clear reasons need to be given for selecting those three regions. E.g., why they represent crop yield at different kinds of background conditions? How do you define those background or climate conditions?

[Response]: Thank you very much for your suggestion. (1) We remove the explanation of "sampling points" in the title of the figure 1. The three selected regions in this study were chosen for comparison with other yield datasets based on different wheatland coverages. Region 1, 2, and 3 represent areas with winter wheat coverages below 25%, around 50%, and above 75%, respectively, serving as representative regions for these respective coverage levels. Modification is incorporated in Fig.1 and Line 136-139.

9) Section 2.2.2, top of atmosphere or surface reflectance was used? Any data quality controls were applied? Please clarify.

[Response]: The GEE platform stores atmospherically corrected reflectance data from Sentinel 2 and Landsat 8, which is the dataset we used (Figure 3). In order to provide a clearer understanding of our dataset to the readers, corresponding modifications have been made in Line 152-155, as follows: "In this work, we extracted the atmospherically corrected reflectance from Landsat 8 and Sentinel 2 images on the Google Earth Engine (GEE) platform during the period of 2016-2021. Subsequently, we calculated the Enhanced Vegetation Index 2 (EVI2) (Jiang et al., 2008) using the extracted reflectance values".

[Figure]

Figure 3 Reflectance dataset from Landsat8 and Sentinel2 based on the GEE platform.

10) Section 2.2.3, is there any high-resolution meteorological data in China? why not use those high-resolution data rather than ERA5?

[Response]: Thanks for your suggestions. (1) ECMWF atmospheric reanalysis refers to data assimilation of near-surface observation data to obtain raster data, which will cover a larger area. This reanalysis combines observations into globally consistent fields taking into account the dynamics and physics of the model using a data assimilation process (four-dimensional variational analysis, 4D-Var, in the case of ERA5). (2) Figure 1 shows the distribution of weather stations in the Huang-Huai-Hai region. It is clear that the existing weather data cannot cover every county. In the process of data analysis, we compared the difference between the weather station data and ECMWF weather data (Figure 2). The results showed that the ECMWF data has a high consistency with the measured data. Similar phenomena occur in other regions. Therefore, ECMWF is a reasonable analysis data set. (3) In our previous research, we have also demonstrated the stability of applying ECMWF data in vegetation monitoring and yield prediction (Xu et al., 2020; Li et al., 2020; Zhao et al., 2022). (4) ERA5 data can be accessed on the GEE platform, which provides convenience for subsequent regional-scale yield prediction. Based on the above, we have chosen to use ERA5 data for subsequent analysis.

[Figure]

Figure 1 Distribution of Meteorological Stations in Huang-Huai-Hai region

[Figure]

Figure 2 Comparison of ECMWF weather data with measured data from weather stations: (a) monthly accumulated temperature ( ℃ ), (b) monthly accumulated precipitation (mm) and (c) monthly accumulated radiation (MJ m-2).

*Xu, X., Teng, C., Zhao, Y., Du, Y., Zhao, C., Yang, G., Jin, X., Song, X., Gu, X., Casa, R., Chen, L., Li, Z.: Prediction of wheat grain protein by coupling multisource remote sensing imagery and ECMWF data. Remote Sensing, 2020, 12: 1349.*

*Li, Z., Taylor, J., Yang, H., Casa, R., Jin, X., Li, Z., Song, X., Yang, G. A hierarchical interannual wheat yield and grain protein prediction model using spectral vegetative indices and meteorological data. Field Crop Res 2020, 248, 107711, https://doi.org/10.1016/j.fcr.2019.107711.*

*Zhao, Y., Han, S., Meng, Y., Feng, H., Li, Z., Chen, J., Song, X., Zhu, Y., Yang, G. Transfer-Learning-Based Approach for Yield Prediction of Winter Wheat from Planet Data and SAFY Model. Remote Sens. 2022, 14, 5474. https://doi.org/ 10.3390/rs14215474*

11) Line 226-227, how did you calculate the average yield using ChinaWheatYiled30m? was it calculated by first summarizing the yield values of pixels within the study area, and then divide by the number of pixels?

[Response]: Thanks for your suggestions. The provincial and municipal average yields based on the ChinaWheatYield30m dataset were calculated by dividing the total yield of all winter wheat pixels by the number of winter wheat pixels in that area. Modification is incorporated in Line 241-243.

12) Fig. 3, what does each point represent? Please clarify it in the figure caption

[Response]: Thank you very much for your suggestion. In order to better display the results, Figure 3 has been changed to a histogram. Modification is incorporated in Fig.3.